# Markov state models of proton- and pore-dependent activation in a pentameric ligand-gated ion channel

Cathrine Bergh[1], Stephanie A Heusser[2†], Rebecca Howard[2], Erik Lindahl[1,2*]

[1]Science for Life Laboratory and Swedish e-Science Research Center, Department of Applied Physics, KTH Royal Institute of Technology, Solna, Sweden; [2]Science for Life Laboratory, Department of Biochemistry and Biophysics, Stockholm University, Solna, Sweden

**Abstract** Ligand-gated ion channels conduct currents in response to chemical stimuli, mediating electrochemical signaling in neurons and other excitable cells. For many channels, the details of gating remain unclear, partly due to limited structural data and simulation timescales. Here, we used enhanced sampling to simulate the pH-gated channel GLIC, and construct Markov state models (MSMs) of gating. Consistent with new functional recordings, we report in oocytes, our analysis revealed differential effects of protonation and mutation on free-energy wells. Clustering of closed- versus open-like states enabled estimation of open probabilities and transition rates, while higher-order clustering affirmed conformational trends in gating. Furthermore, our models uncovered state- and protonation-dependent symmetrization. This demonstrates the applicability of MSMs to map energetic and conformational transitions between ion-channel functional states, and how they reproduce shifts upon activation or mutation, with implications for modeling neuronal function and developing state-selective drugs.

*For correspondence:
erik.lindahl@dbb.su.se

Present address: †Department of Drug Design and Pharmacology, University of Copenhagen, Copenhagen, Denmark

Competing interests: The authors declare that no competing interests exist.

## Introduction

The family of pentameric ligand-gated ion channels (pLGICs), also known as Cys-loop receptors, controls electrochemical signal transduction in numerous tissues and cell types, from bacteria to humans. A rapid cycling between conducting and nonconducting conformations in response to chemical stimuli, such as neurotransmitter binding or changes in pH, is fundamental to their function. These channels are often found in the postsynaptic membrane of neurons and undergo allosteric conformational changes, where the pore in the integral transmembrane domain (TMD) opens for ion conduction upon selective neurotransmitter binding in the extracellular domain (ECD). Prokaryotic homologs, such as the proton-gated channel GLIC from the cyanobacterium *Gloeobacter violaceus*, share many topological features with eukaryotic pLGICs, and have been proposed to follow similar gating pathways (*Figure 1A–C*). For any pLGIC, channel function can ultimately be explained by its free energy landscape, where different state populations shift upon activation through agonist binding. Understanding these landscapes and how the free energies either of stable states or barriers change upon ligand-binding or mutation is thus crucial for full understanding of the gating process, with applications including the development of state-dependent drugs for better treatment of diseases related to channel malfunction.

Recent advances in structural biology have enabled a steady increase in the number of available pLGIC structures. The GLIC model system is notable in this regard, accounting for over 40% of pLGIC entries in the protein data bank, including apparent closed and open states. However, the conformational diversity of these states is highly limited, leading to crude representations of the free-energy landscapes from experimental structures alone. Computational methods like molecular

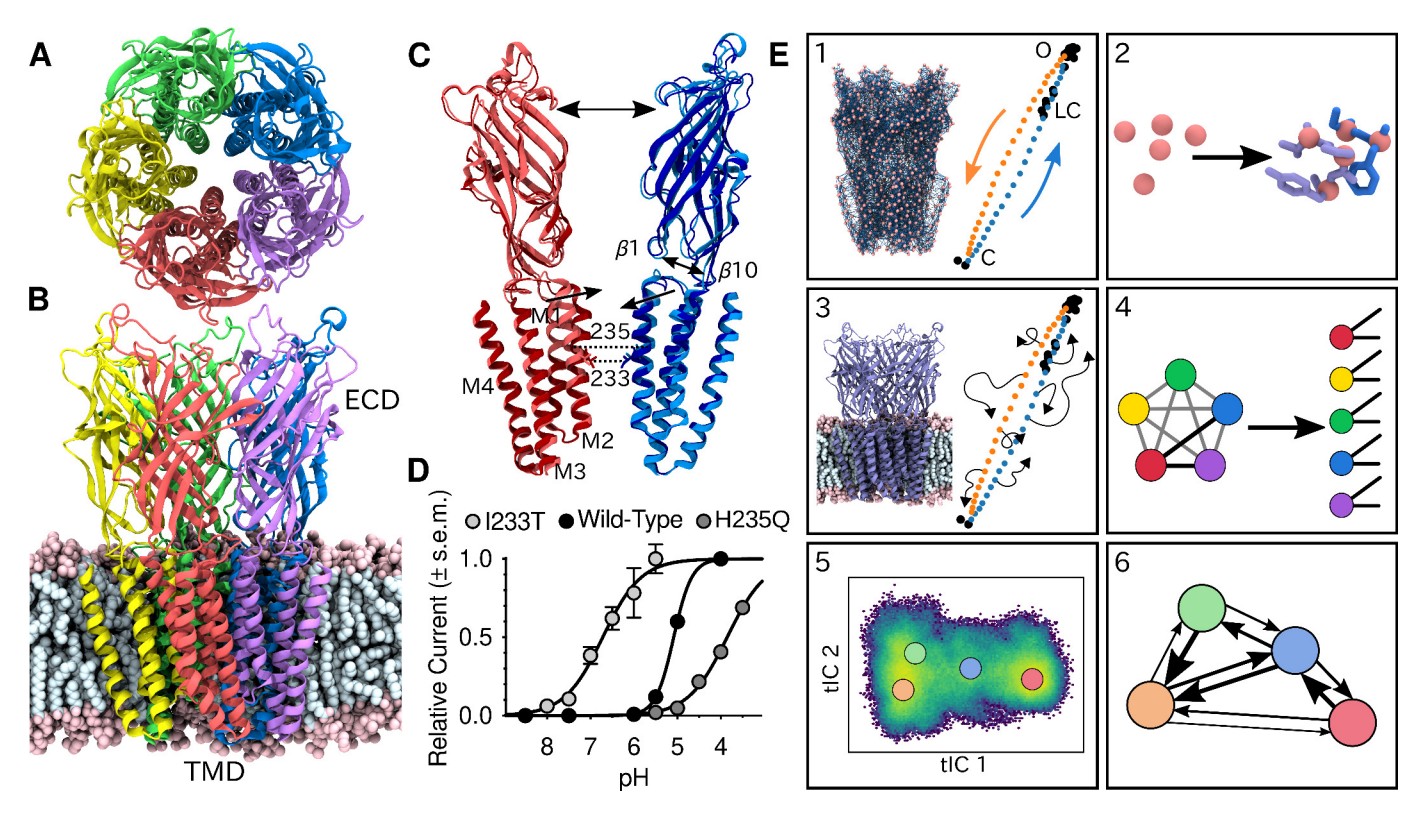

**Figure 1.** Global architecture of GLIC, electrophysiology data and computational methodology. GLIC in an open conformation shown from (**A**) the top and (**B**) the side in a POPC lipid bilayer. (**C**) Two opposing subunits highlighting the pore of the channel. Light colors represent the open conformation (PDB ID 4HFI) and dark colors the closed one (PDB ID 4NPQ). Arrows indicate important gating motions - the tilting of the M2 helices, beta expansion and ECD spread. Residues I233 and H235 on the pore-lining M2 helices were mutated in both simulations and electrophysiology experiments. (**D**) Electrophysiology data for wild-type GLIC, the gain-of-function I233T variant and the loss-of-function H235Q variant. (**E**) Simulation methodology: the eBDIMS method provides (1) coarse-grained seed structures along the transition pathway followed by (2) reconstruction of the atomistic detail. Atomistic structures were then (3) embedded in lipid bilayers and massively parallel unrestrained MD launched. Analysis involves (4) a feature transformation to account for the symmetry of the pentamer, followed by (5) dimensionality reduction with tICA and (6) MSM construction.

dynamics (MD) simulations can be used to sample more of the conformational landscape, and several studies have been conducted on GLIC to study short-timescale motions; such as simulations of the transmembrane domain only (*Zhu and Hummer, 2010*; *Zhu and Hummer, 2012a*; *Zhu and Hummer, 2012b*), studies of the ion permeation pathway through potential-of-mean-force calculations (*Cheng et al., 2010*; *Fritsch et al., 2011*), and steady-state simulations reaching 100 ns to 1 µs timescales (*Nury et al., 2010*; *Prevost et al., 2012*; *Calimet et al., 2013*), some also with additional ligands or modulations (*Brannigan et al., 2010*; *Willenbring et al., 2011*; *LeBard et al., 2012*; *Murail et al., 2012*; *Laurent et al., 2016*; *Heusser et al., 2018*; *Faulkner and de Leeuw, 2020*). Still, due to the large system size and relatively long timescales of the gating transitions, in practice it has not been feasible to sample complete gating transitions, especially if ligand-binding and unbinding events are involved (*Chakrapani and Auerbach, 2005*; *Velisetty and Chakrapani, 2012*; *Gonzalez-Gutierrez et al., 2012*; *Laha et al., 2013*; *Gonzalez-Gutierrez et al., 2013*; *Velisetty et al., 2012*; *Menny et al., 2017*). To bridge this gap, various enhanced sampling methods can be used, often by the application of biasing forces or presumed reaction coordinates. For instance, application of the string method with swarms of trajectories recently enabled the identification of local rearrangements in channel closure, including contraction of the upper pore, loosening of β-strand contacts in the lower ECD, and general expansion of the upper ECD (*Lev et al., 2017*). This provides precious information of structural rearrangements, but the choice of collective variables in combination with the timescales of individual simulations may influence what motions are sampled.

Markov state models (MSMs) show promise in modeling the thermodynamics and kinetics of biological systems without making prior mechanistic assumptions (*Dämgen and Biggin, 2019*; *Prinz et al., 2011*; *Husic and Pande, 2018*). By counting transitions, multiple simulations can effectively be stitched together to capture processes on timescales longer than any individual simulation, and probing more biologically relevant dynamics. However, in principle MSMs first require the whole equilibrium distribution to be sampled, which is difficult to achieve by starting simulations from experimental structures alone, since the stability required for crystal packing or cryo-EM data processing results in structures sometimes representing metastable states. Simulations started from these states then usually remain confined to the energy well for long time periods. In contrast, seeding simulations at regions in the free energy landscape that are not necessarily metastable enhance sampling without introducing any biasing forces, and the actual sampling is performed without limiting the system to any particular reaction coordinate.

Here, we have used such enhanced seeding approaches combined with MSMs to sample the GLIC opening-closing transition. Both the wild-type and two different variants with mutations along the pore-lining M2 helices - where we also showed one to yield gain-of-function similar to human homologs (*Filatov and White, 1995*; *Kosolapov et al., 2000*) and the other loss of function (*Fourati et al., 2018*) - were simulated in resting or activating conditions (neutral or low pH). From the resulting trajectories we were able to construct MSMs of the complete ion channel, quantitatively map free-energy landscapes, extract molecular details regarding the gating mechanisms, and show how the gating shift observed in electrophysiology recordings of the wild-type and variants is modeled correctly by the MSMs. Additionally, we present new evidence for the role of symmetry in gating.

## Results

### Enhanced sampling enables MSM construction of GLIC gating

To shed light on the gating mechanisms, we combined enhanced sampling of MD simulations at resting and activating pH with Markov state modeling. X-ray structures of GLIC crystallized at pH 7 (PDB ID 4NPQ) and pH 4 (PDB ID 4HFI) have been reported to represent closed and open states, respectively (*Sauguet et al., 2014*; *Figure 1C*). Because unbiased molecular dynamics simulations started from these states alone were not expected to thoroughly sample the activation process, we seeded simulations along the presumed gating transition which enabled simulations to run in a massively parallel fashion. To achieve this, the starting structures were simplified to Cα traces, and used to drive elastic network-driven Brownian dynamics (eBDIMS) (*Orellana et al., 2016*), where the protein, represented as an elastic network, was pushed from closed to open X-ray structures and vice versa to generate two initial pathways (*Figure 1E1*). Following side-chain reconstruction (*Figure 1E2*), this approach resulted in a set of 50 initial models interspersed in principal component space between the open and closed X-ray structure clusters, all with standard titration states representing neutral pH (deprotonated) that in experiments should result in a closed channel. In a duplicate set of initial models, a subset of acidic residues was modified to the most probable titration state under activating conditions (protonated), as previously described (*Nury et al., 2011*). Each resulting model was then subjected to unrestrained simulation in a palmitoyloleoylphosphatidylcholine (POPC) lipid bilayer for over 1 μs, producing 120 μs total sampling in the two conditions (*Figure 1E3*, *Figure 2—figure supplement 1*). Among simulations performed at each pH, we performed a feature transformation to account for the symmetry of the homopentameric protein (*Figure 1E4*), followed by dimensionality reduction by time-lagged independent component analysis (tICA) (*Pérez-Hernández et al., 2013*; *Schwantes and Pande, 2013*), to capture the slowest motions observed in the simulations (*Figure 1E5*). Further clustering in the resulting tICA space yielded kinetically meaningful states that could be used for MSM construction (*Figure 1E6*), validated by convergence of the main transition timescale along with other statistical tests of eigenvalues, eigenvectors and sampling (*Figure 2—figure supplement 2*; *Figure 2—figure supplement 3*).

To assess the ability of this computational approach to predict functional properties, we introduced the gain-of-function mutation I233T, located at the midpoint of the GLIC transmembrane pore, as well as the loss-of-function mutation, H235Q, a few residues away facing the intrasubunit helical bundle (*Figure 1C*). The midpoint position, 9' counting from the intracellular side, has been

shown to constitute a hydrophobic gate that critically influences conduction properties in GLIC as in other family members (*Gonzalez-Gutierrez et al., 2017*), while substitution at the interfacial H235, or 11' position, enabled crystallization of an intermediate, locally closed conformation (*Prevost et al., 2012*). Indeed, we confirmed by two-electrode voltage-clamp electrophysiology recordings in *Xenopus oocytes* that the I233T substitution enhanced proton sensitivity, shifting half-maximal activation by more than one pH unit, and producing moderately conductive channels even at neutral pH. Conversely, the histidine substitution decreased proton sensitivity, resulting in moderately conductive channels at low pH (*Figure 1D*). Note that maximal conduction in this plot does not necessarily indicate that all channels are open. These I233T and H235Q substitutions were introduced into additional replicate sets of initial simulation models in both deprotonated and protonated states as described above. The resulting mutants were prepared, simulated, transformed, and analyzed in the same way as the wild-type channels, achieving convergence on similar timescales (*Figure 2—figure supplement 2A,D*). Thus, we were able to produce six independent, statistically validated, MSMs representing distinct combinations of pH and gain- or loss-of-function mutant variations.

## Free-energy landscapes capture effects of protonation and mutation

Free-energy landscapes obtained from the first eigenvector of each MSM clearly distinguished closed- and open-like regions in all six conditions (*Figure 2*). The tICA coordinates are presumed to represent the two slowest motions, where the largest eigenvector contributions are originating primarily from interactions between helices in the transmembrane domain (*Figure 2—figure supplement 4*). Along the first tIC – with the largest eigenvector contributions focused around the M2 helices (*Figure 2—figure supplement 4A,C,E*) – the closed X-ray structure consistently projected to the lowest free energy minimum in the landscape, suggesting a predominant population of closed channels under both deprotonated and protonated conditions. Due, in part, to its low conductance in single-channel recordings (*Bocquet et al., 2007*), the open probability of GLIC is not well established; however, recent cryo-electron microscopy studies support a predominance of nonconducting states even at pH below 4 (*Rovšnik et al., 2021*). In deprotonated conditions, the I233T substitution created a well-defined second local minimum in free energy along tIC1, centered near the open X-ray structures (*Figure 2C*), not seen in the wild-type or H235Q variants (*Figure 2A,E*). Protonated conditions further deepened this secondary minimum for both wild-type and I233T variants (*Figure 2B,D*), but for the H235Q variant this secondary minimum was notably displaced along tIC2 such that it did not overlap with the open X-ray structures (*Figure 2F*). Despite topological differences, the distribution of closed versus open X-ray structures was comparable along tIC1 in all six conditions, suggesting a common component for this slowest transition. Several so-called locally closed structures (e.g. PDB ID 3TLS) – featuring a closed-like TMD but open-like ECD – projected to a region intermediate along tIC1, but within the broad closed-state free energy well, suggesting this component primarily distinguishes TMD rather than ECD state. Interestingly, a few locally closed structures were separated out along tIC2 into a shallow secondary minimum of the closed state energy well for the deprotonated H235Q variant (*Figure 2E*). Conversely, a lipid-bound state (PDB ID 5J0Z) - in which the pore is expanded at the outward vestibule, although nonconductive at the hydrophobic gate - clustered with open channels. In contrast to tIC1, tIC2 did not distinguish X-ray structures in a consistent manner; the largest eigenvector contributions were interspersed between multiple transmembrane helices (*Figure 2—figure supplement 4B,D,F*) and higher values of this component may represent conformations sampled primarily in MD simulations.

## Kinetic clustering distinguishes metastable open and closed states

To quantify population shifts in the evident closed- and open-like regions of the tIC landscape, we coarse-grained the MSMs into separate metastable states. By clustering according to the first dynamical MSM eigenvector, each free energy landscape could be divided into two macrostates corresponding to closed and open conformations, respectively (*Figure 3A*). In all but the deprotonated wild-type and H235Q landscapes, an evident free energy barrier indicated these states to be metastable (*Figure 2*). Based on this clustering, and consistent with the qualitative comparisons above, the fractional population of open-like macrostates moderately increased (6% to 12%) upon I233T substitution under deprotonated conditions, and increased further for both variants upon

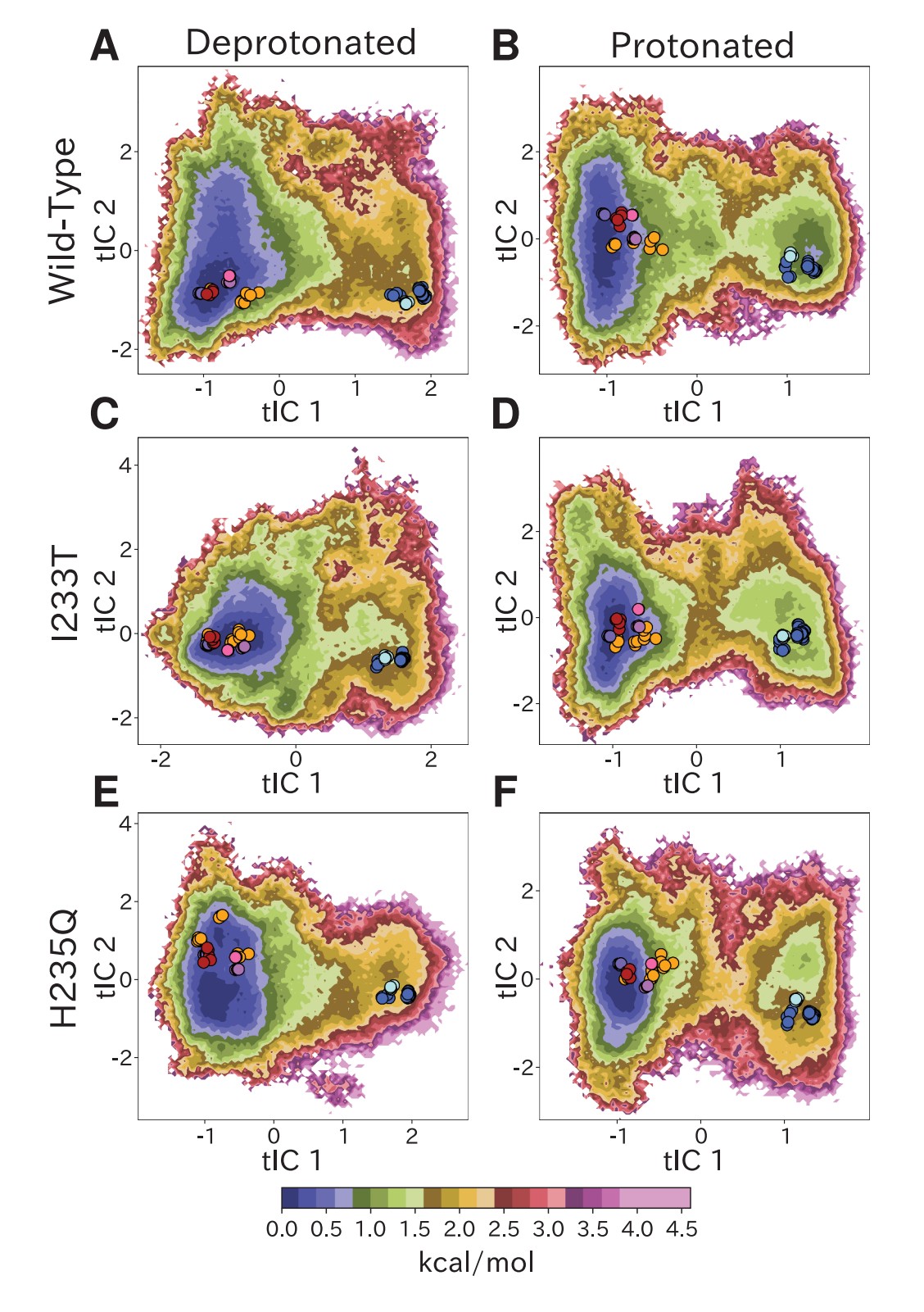

**Figure 2.** Free-energy landscapes capture shifts upon protonation and mutation. Free-energy landscapes projected onto the first two tICA coordinates for (A) deprotonated wild-type, (B) protonated wild-type, (C) deprotonated I233T mutant, (D) protonated I233T mutant, (E) deprotonated H235Q mutant, and (F) protonated H235Q mutant. Dots indicate experimental structures, with red representing closed X-ray structures at pH 7 (PDB ID 4NPQ), orange locally closed X-ray structures (PDB IDs 3TLS, 5MUO, 4NPP(B)), light blue modulated states (PDB ID 5J0Z) and blue open states (PDB IDs 4HFI, *Figure 2 continued on next page*

*Figure 2 continued*

3EAM, 3P4W, 4IL4, 3UU5, 4NPP(A)). Pink, magenta, and purple dots indicate closed cryo-EM structures at pH 7, 5, and 3 (PDB IDs 6ZGD, GZGJ, GZGK) respectively. Less than 100% of the channels are expected to adopt an open state even under protonated conditions (**B, D, F**). At protonated conditions the H235Q mutation results in an 'open'-state free-energy minimum distinct from experimental open structures (**F**).

The online version of this article includes the following figure supplement(s) for figure 2:

**Figure supplement 1.** Simulations sample the initial transition pathway broadly.

**Figure supplement 2.** Assessment of convergence and sampling of the MSMs.

**Figure supplement 3.** Variational optimization for selection of hyperparameters.

**Figure supplement 4.** Vector representation of the two tICA coordinates.

protonation (to 17% and 20% for wild-type and I233T, respectively). Upon H235Q substitution the open state population instead decreased slightly (6% to 3%) under deprotonated conditions, and reached a lower open probability (15%) compared to wild-type after protonation (**Figure 3B**). MSM kinetics indicated the I233T mutation accelerated opening transitions, decreasing the closed–open mean first-passage time by over one third (8.9 to 5.8 μs). Conversely, the H235Q mutation substantially slowed opening transitions by almost doubling the transition times (8.9 to 13.3 μs and 8.0 to 14.3 μs). Additionally, protonation appeared to slow closing transitions, increasing open–closed times nearly 3-fold for wild-type and I233T variants (0.5 to 1.4 μs and 0.6 to 1.7 μs for wild-type and I233T, respectively), while increasing five-fold for the H235Q mutant (0.4 to 2.1 μs) (**Figure 3C**).

To further validate the functional annotation of closed- and open-like macrostates, we plotted pore hydration across each free-energy landscape (**Figure 3D**) and extracted populations of hydrated conformations (**Figure 3B**). Under all conditions, the macrostate barrier corresponded to a dramatic shift in hydration levels along tIC1. All regions of tIC space sampled some dehydrated states, likely corresponding to transient, reversible, obstructions observed in individual trajectories (**Figure 3—figure supplement 1**; **Figure 3—figure supplement 2**; **Figure 3—figure supplement 3**). However, protonation increased hydration for all variants, particularly at larger (open-like) values of tIC1 (**Figure 3D**). As expected with a polar residue at the hydrophobic gate, the I233T substitution further increased hydration at all values along tIC1, suggesting more closed-like states might achieve ion conduction in this variant. Conversely, in addition to being displaced from experimental structures (**Figure 2F**), the open-like free-energy well was substantially less hydrated in the context of H235Q, indicating a reduced propensity for ion conduction. Thus, qualitative and quantitative comparisons of tIC landscapes supported reproducible state distinctions, and recapitulated functional effects of both protonation and mutation, supporting our model as a reasonable representation of GLIC gating.

## Higher order clustering reveals conformational trends in gating

To identify conformations along the gating pathway more precisely, we reclustered each dataset according to a larger eigenvector set, obtaining models with four or five states each (**Figure 4A**). Note that these states are not necessarily metastable, but this clustering enables further studies of the different kinetically similar regions of the energy landscapes. Conformations corresponding to low values of both tIC1 and tIC2 (state I, **Figure 4A**) consistently comprised the most populated cluster (**Figure 4B**), with representative samples featuring visibly tilted M2 helices and a contracted pore (doi:10.5281/zenodo.5500174). Accordingly, closed as well as locally closed X-ray structures projected to the state-I cluster in all but the protonated wild-type and deprotonated H235Q systems (**Figure 4A**). Two more states with contracted pores clustered at low–intermediate values of tIC1, varying somewhat with system conditions; these states were distinguished along tIC2 (high for state II, low for state III; **Figure 4A**). Representative conformations in state II were visibly similar to state I (doi:10.5281/zenodo.5500174). For the protonated wild-type system, the closed X-ray structure projected in State II, with locally closed structures along the state I–II border. For the deprotonated H235Q variant, locally closed structures (including the experimental structure of this variant) clustered in state II, with closed states near the state I-II border (**Figure 4A**). State III corresponded visibly to a partly expanded pore and often high degrees of TMD subunit asymmetry (doi:10.5281/zenodo.5500174), not represented by any known X-ray structures (**Figure 4A**). Open and lipid-modulated X-ray structures projected to a cluster at high tIC1 and low tIC2 (state IV, **Figure 4A**),

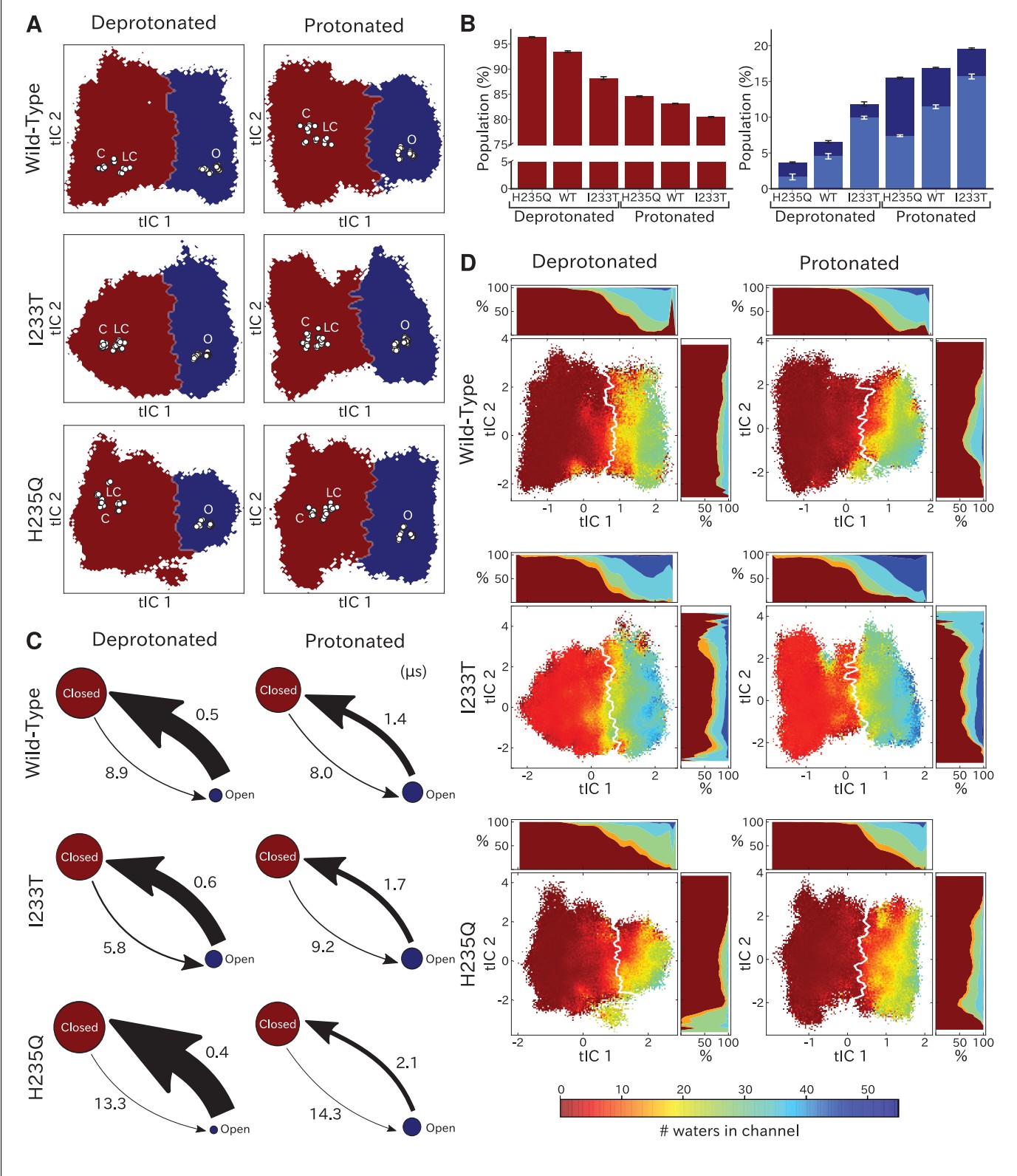

**Figure 3.** Two-state clustering distinguishes metastable open and closed states. (**A**) Two metastable states separated by the highest free energy barrier, with red representing closed-like states and blue open-like states. White dots indicate experimental structures, with labels marking closed (**C**), locally closed (**LC**) and open (**O**) clusters. (**B**) Populations of the closed (red) and open (dark blue) macrostates, with populations of hydrated conformations (>20 water molecules in the pore) marked in lighter blue. Protonated conditions consistently decreased closed-like and

*Figure 3 continued on next page*

*Figure 3 continued*

increased open-like populations, albeit to less than 100%. Relative to wild-type, the I233T substitution decreased closed-like populations, while H235Q decreased open-like populations, and even more so the population of hydrated conformations. Relative changes in open population reflect shifts in functional activity in *Figure 1*. (C) Transition rates (arrows) with numbers representing mean first passage times of crossing the highest free energy barrier. Transition rates to the closed state consistently decreased under protonated versus deprotonated conditions. In deprotonated conditions, the I233T mutation increased the rate of opening, while the H235Q mutation decreased opening rates in both conditions. (D) Hydration of the transmembrane pore. Side panels show the population of states with different hydration levels. The pore can be seen hydrating or dehydrating when crossing the main free energy barrier (white). The I233T mutation resulted in higher levels of hydration in both open and closed states, while the H235Q mutation was less hydrated in the open state.

The online version of this article includes the following figure supplement(s) for figure 3:

**Figure supplement 1.** Pore hydration in wild-type GLIC simulations.
**Figure supplement 2.** Pore hydration in I233T mutant GLIC simulations.
**Figure supplement 3.** Pore hydration in H235Q mutant GLIC simulations.

substantially populated only upon protonation and/or I233T substitution (*Figure 4B*); as expected, representative samples featured the most expanded pore relative to other states (doi:10.5281/zenodo.5500174). A final cluster at high values of both tICs (state V, *Figure 4A*) was the least populated in all conditions, except for the protonated H235Q variant where the secondary free energy minimum had shifted to state V rather than state IV (*Figure 4B*), again sampling intermediate values of pore expansion. Visual inspection of all macrostates (*Figure 4A*) and free-energy landscapes (*Figure 2*) indicated that the most likely transition pathway between closed- (states I/II) to open-like (state IV) would proceed via state III, or state II in the protonated H235Q model.

To further validate these multi-state spaces in context of past mechanistic models, we compared our clusters on the basis of conformational features implicated in channel gating.

As previously described, a key distinction between closed- and open-like states was expansion of the upper-TMD pore, quantified here by the radial distance of the upper M2 helix from the pore center-of-mass (M2 spread). On this basis, states I/II and IV/V were respectively contracted and expanded, with state III – and state V in the protonated H235Q variant – sampling intermediate values (*Figure 5A*; *Figure 5—figure supplement 3*). The expansion around the main hydrophobic gate at the 9′ position was generally expanded in the I233T mutant versus wild-type conditions (*Figure 5B*; *Figure 5—figure supplement 4*), as expected upon substitution of threonine for isoleucine. However, the H235Q mutant instead resulted in a more constricted pore with the most populated state V values approaching those of wild-type closed-state. All states were somewhat constricted at 9′ relative to the open X-ray structure (*Figure 5B*), though our previous measurements confirmed sampling of hydrated conformations ($\geq$30 water molecules in the channel pore, *Figure 3D*) in open-like states in both conditions. Interestingly, a proposed secondary gate at the intracellular end of the pore (−2′ radius, *Figure 5—figure supplement 1A*; *Figure 5—figure supplement 9*) implicated in desensitization (*Gielen and Corringer, 2018*) exhibited a bimodal distribution in radii, with state I largely retaining an expanded $\geq$5-Å radius similar to X-ray structures, while other states partially sampled a contracted radius ~2 Å.

In parallel with expansion of the upper pore, gating transitions have been associated with contraction at TMD subunit interfaces, quantified here by the distance between proximal regions of the principal upper-M2 and complementary upper-M1 helices (M2$^+$-M1$^-$ distance). Subunit interfaces in states I/II and IV/V were expanded and contracted respectively, with state III again sampling intermediate values (*Figure 5C*; *Figure 5—figure supplement 5*). Transitions at this interface have also been linked to relief of a helix kink, proximal to a conserved proline residue in M1. Indeed, this helix kink was more acute in states I/II, sampling even smaller angles than closed X-ray structures; conversely, the kink was largely relieved in state IV, with even larger angles than open structures (*Figure 5—figure supplement 1B*; *Figure 5—figure supplement 10*). State III again sampled intermediate values. Interestingly, kink angles for state V overlapped states I/II in deprotonated conditions, but shifted toward state IV in protonated conditions.

Gating transitions in the TMD are coupled to allosteric rearrangements in the ECD, particularly so-called β-expansion involving the first and last extracellular β-strands in each subunit. Proximal to the TMD interface, the cleft between these ECD strands is relatively expanded in closed X-ray structures but contracted in open structures, strengthening a salt bridge between β1-D32 and β10-R192

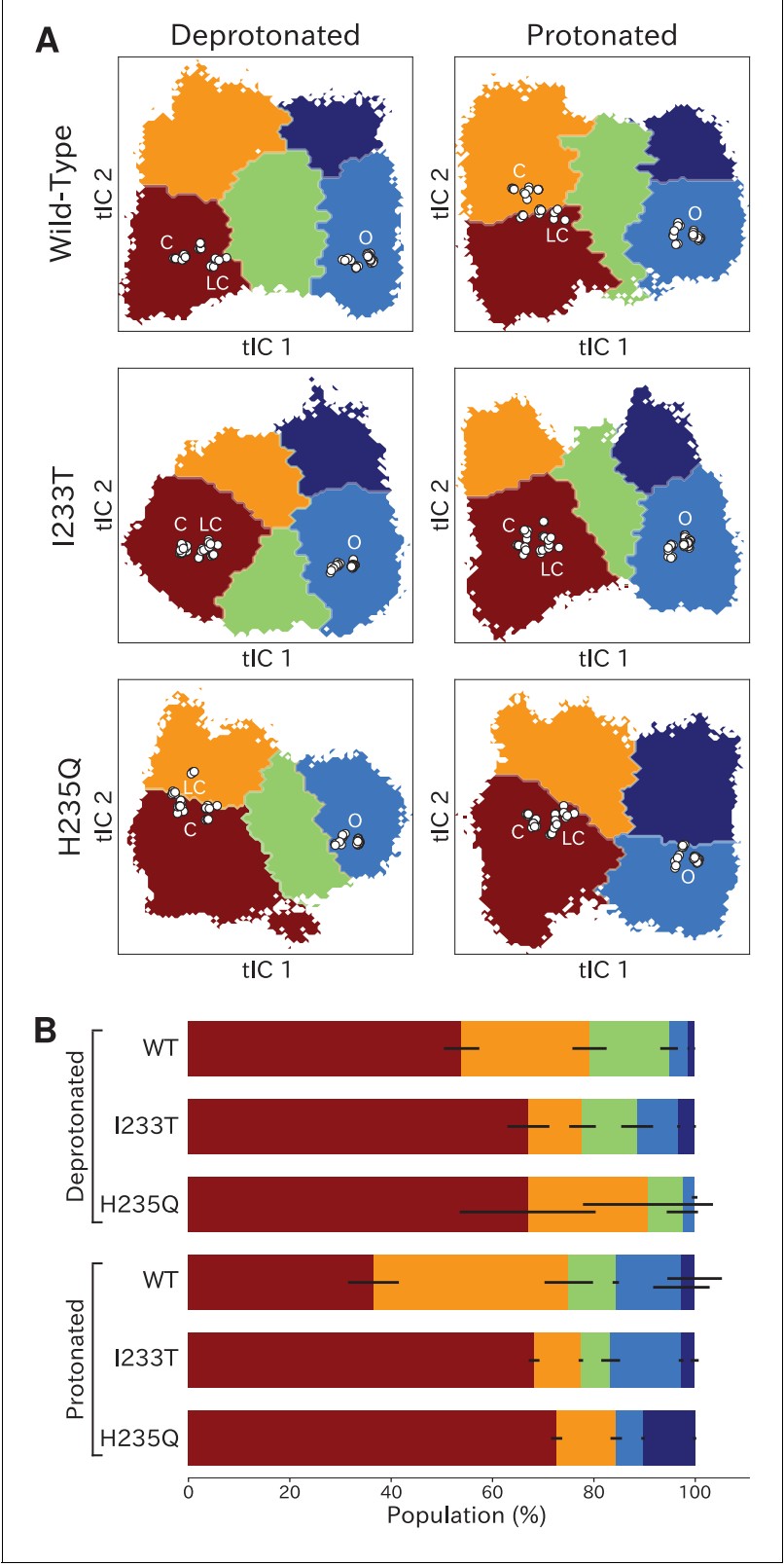

**Figure 4.** Higher order clustering of the GLIC free-energy landscapes. (**A**) Each free-energy landscape can be further clustered into models with four or five macrostates that – despite not being metastable– allow for more fine-grained structural analysis of the energy landscape. White dots represent experimental structures marking closed (**C**), locally closed (**LC**) and open (**O**) clusters, respectively. (**B**) Populations for each macrostate. The different

*Figure 4 continued on next page*

Figure 4 continued

regions will be referred to as: red - State I, orange - State II, green - State III, light blue - State IV, and dark blue - State V. Conformations sampled from these states can be accessed at doi:10.5281/zenodo.5500174.

(*Lev et al., 2017*). Interestingly, values of β-expansion exhibited a bimodal distribution, with distinct peaks centered around closed and open structures (*Figure 5D*; *Figure 5—figure supplement 6*). In our MSMs, state-I samples were more expanded, while states IV/V were more contracted; distributions in states II/III featured distinct peaks at both closed- and open-like values.

More global gating motions in the ECD are thought to include blooming and twisting, with channel activation involving contraction and untwisting relative to the TMD. Surprisingly, no consistent state-dependent trends were noted in ECD spread (*Figure 5E*; *Figure 5—figure supplement 7*) or twist (*Figure 5—figure supplement 1C*; *Figure 5—figure supplement 11*). However, comparing for example the predominating state I in each condition revealed an overall pH dependence, with the ECD generally more contracted and twisted in protonated than deprotonated conditions (*Figure 5E*; *Figure 5—figure supplement 1C*). Time-series of ECD blooming and twisting motions from all trajectories showed rapid adaptation in response to change in pH regardless of initial seed ECD conformation (*Figure 5—figure supplement 2*). However, simulations started in an overall closed-like conformation displayed more flexible ECDs and sampled more broadly than simulations started in an open-like conformation, indicating state-dependent differences in ECD flexibility rather than average ECD blooming or twisting values. Interestingly, the ECD rarely contracted to the extent of open X-ray structures, nor twisted to the extent of closed X-ray structures in any condition, suggesting that crystal contacts may favor uncommon conformations in this domain. A recent cryo-EM structure of the closed state at pH 3 (*Rovšnik et al., 2021*) suggest a more compact ECD compared to the closed-state X-ray structure which align better with the expectation value of our data at low pH (*Figure 5E*), further supporting the idea that crystal contact might favor compact ECDs in the open state X-ray structure.

Finally, we studied E35, thought to represent the main proton sensor in GLIC (*Nemecz et al., 2017*; *Hu et al., 2018*). The probability distributions of the distance between E35 and T158 of the complementary subunit show clear shifts in response to changes in protonation, leading to local backbone rearrangements (*Figure 5F*; *Figure 5—figure supplement 8*). Indeed, our simulations sampled even tighter contacts between these residues than observed in experimental structures. Thus, our models capture pH-dependent changes around E35, further supporting the role of E35 as an important proton sensor.

## Symmetry analysis reveal protonation- and state-dependent differences

Although each GLIC molecule is composed of five identical subunits and exhibits fivefold symmetry in the context of a crystal, it remains unclear how symmetry is retained or broken in the course of channel gating. Previous simulations suggested a role for conformational asymmetry, particularly in the TMD, in facilitating closing transitions (*Nury et al., 2010*; *Mowrey et al., 2013b*). To estimate subunit symmetry, we quantified pairwise RMSDs between homologous atoms in neighboring and opposing subunits, and plotted the resulting symmetry value for each simulation frame to its corresponding position in tIC space. In this representation, regions of higher pairwise RMSD correspond to lower symmetry. To enable identification of domain-specific changes, analyses were also performed independently for the TMD and ECD in each condition (*Figure 6*, top bars). In both protonated datasets, pairwise RMSDs of the open state were significantly lower compared to deprotonated conditions, suggesting that protonation plays a role in the symmetrization of this state. These high symmetry levels could be deduced from both the ECD and TMD (*Figure 6B, D and F*, top bars). Upon channel closure, the overall symmetry decreases, which is primarily driven by the symmetry loss seen in the ECD. The TMD on the other hand seems to recover some symmetry in the closed state. At deprotonated conditions, pairwise RMSDs were generally higher across the board, and neither ECD or TMD were able to reach the low RMSD levels of the open state at protonated conditions. The ECD symmetry still seemed to decrease upon channel closure, but the differences were diminished compared to protonated conditions (*Figure 6A, C and E*, top bars).

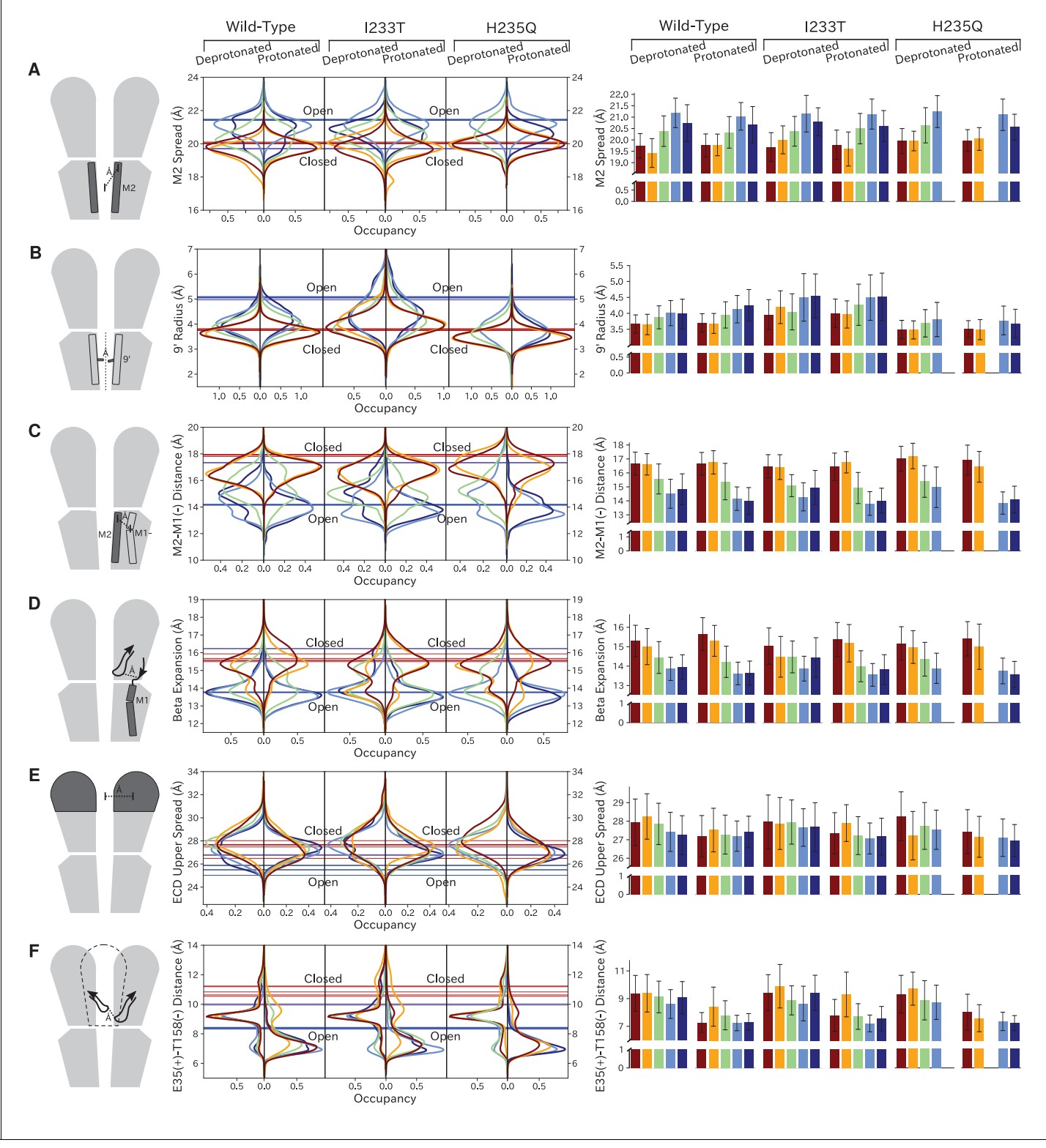

**Figure 5.** Probability distributions of a few variables proposed to be important in GLIC gating. The left-most cartoons illustrate the definition of each variable, while data is presented as probability distributions with means and standard deviations plotted as bars. Colors represent the macrostates in *Figure 4*, and blue, red and purple horizontal lines represent the experimental structures 4HFI (*Sauguet et al., 2013*), 4NPQ (*Sauguet et al., 2014*) and 6ZGK (*Rovšnik et al., 2021*), respectively. The spread of the pore-lining M2 helices (**A**, *Figure 5—figure supplement 3*) is captured by the open and closed macrostates, with intermediate states taking intermediate values. However, the open-state minimum is more contracted for the protonated

*Figure 5 continued on next page*

*Figure 5 continued*

H235Q variant (state V). The radius from the pore center to the 9' hydrophobic gate (B, *Figure 5—figure supplement 4*) captures clear differences between the three variants, where the I233T mutant has a more expanded pore than wild-type while the pore of H235Q is more contracted. The distance between the pore-lining M2 helix and the M1 helix of the neighboring subunit (C, *Figure 5—figure supplement 5*) is also correctly represented by the open and closed macrostates, with State III taking intermediate values. The beta expansion (D, *Figure 5—figure supplement 6*) yields distributions with expectation values of closed and open states aligning well with the experimental structures, while the intermediate state produces a bimodal distribution. Interestingly, the probability distributions of the closed-like states of the I233T mutation at deprotonated conditions show increased biomodality as well. The upper spread of the extracellular domain (ECD) (E, *Figure 5—figure supplement 7*) does not result in a clear separation of the macrostates, but a smaller pH-dependent shift can be observed. The C$\alpha$ distance between E35 and T158 (F, *Figure 5—figure supplement 8*) of primary and complementary subunits, respectively, capture large pH-dependent shifts in for all macrostates.

The online version of this article includes the following figure supplement(s) for figure 5:

**Figure supplement 1.** Distributions of variables proposed to be important in GLIC gating.
**Figure supplement 2.** ECD spread and twist values show rapid adaption in response to pH.
**Figure supplement 3.** M2 spread values projected onto the first two tICs.
**Figure supplement 4.** 9' distances projected onto the first two tICs.
**Figure supplement 5.** M2-M1(-) values projected onto the first two tICs.
**Figure supplement 6.** Beta expansion values projected onto the first two tICs.
**Figure supplement 7.** ECD upper spread values projected onto the first two tICs.
**Figure supplement 8.** The distance between residues E35 and T158 projected onto the first two tICs.
**Figure supplement 9.** -2' distances projected onto the first two tICs.
**Figure supplement 10.** M1 kink values projected onto the first two tICs.
**Figure supplement 11.** ECD twist values projected onto the first two tICs.

## Discussion

We have constructed Markov state models of a pentameric ligand-gated ion channel that enabled quantitative modeling of protonation and mutation effects, identification of intermediate states and characterization of the effect of symmetry in channel gating. Our free-energy landscapes showed deepening of the open state free energy well upon protonation, destabilization of the closed state upon I233T mutation of the hydrophobic gate (*Figure 7A*), and shifting of the open state toward conformations with more constricted and less hydrated pores upon H235Q mutation (*Figure 7B*). Our models captured shifts in free energies in agreement with our electrophysiology recordings, although only a fraction of channels would be open even under protonated conditions. In addition to capturing features of the gating mechanism already proposed for pLGICs, our MSMs allowed for further exploration of features that correlate with gating. Here, we focused on the effect of conformational symmetry between subunits and found that the open state displayed higher levels of symmetry, which was particularly enhanced after protonation (*Figure 7C*).

Thermodynamic properties calculated from the present models were largely consistent with functional recordings, showing a shift toward relative stabilization of open versus closed states upon protonation of acidic residues or polar substitution at the 9' hydrophobic gate (*Figure 7A*). Interestingly, along with a modest decrease in population, the open-like state itself changed to represent conformations with more constricted and less hydrated pores upon H235Q substitution (*Figure 7B*). Free energies of the wild-type gating transition have previously been estimated using string method optimization of a few collective variables assumed to be important in the gating transition (*Lev et al., 2017*). The variable associated with the largest barrier height in the work of Lev et al. is pore hydration, which is an integral part of the gating transition, thus suitable for comparison to the maximal energy barrier height from the MSMs. At low pH the string method hydration yielded a barrier of 1.5 kcal/mol, compared to 1.0–1.5 kcal/mol for the protonated MSM. At neutral pH the barrier from string method hydration gave 2.5 kcal/mol, while the MSM resulted in 1.5–2.0 kcal/mol height of the energy barrier. This indicated that the wild-type MSMs found transition pathways with comparable free-energy barriers.

Two-state clustering suggested a value of $P_{open}$ well below 100% for GLIC, even at activating conditions. Although $P_{open}$ has not been determined for GLIC, recent cryo-EM structures of closed-state GLIC solved at activating conditions point toward a significant closed population even at activating conditions (*Rovšnik et al., 2021*; *Sauguet et al., 2014*). Additionally, recent small-angle neutron scatting (SANS) experiments estimated low-pH $P_{open}$ to 18% by fitting linear combinations of closed

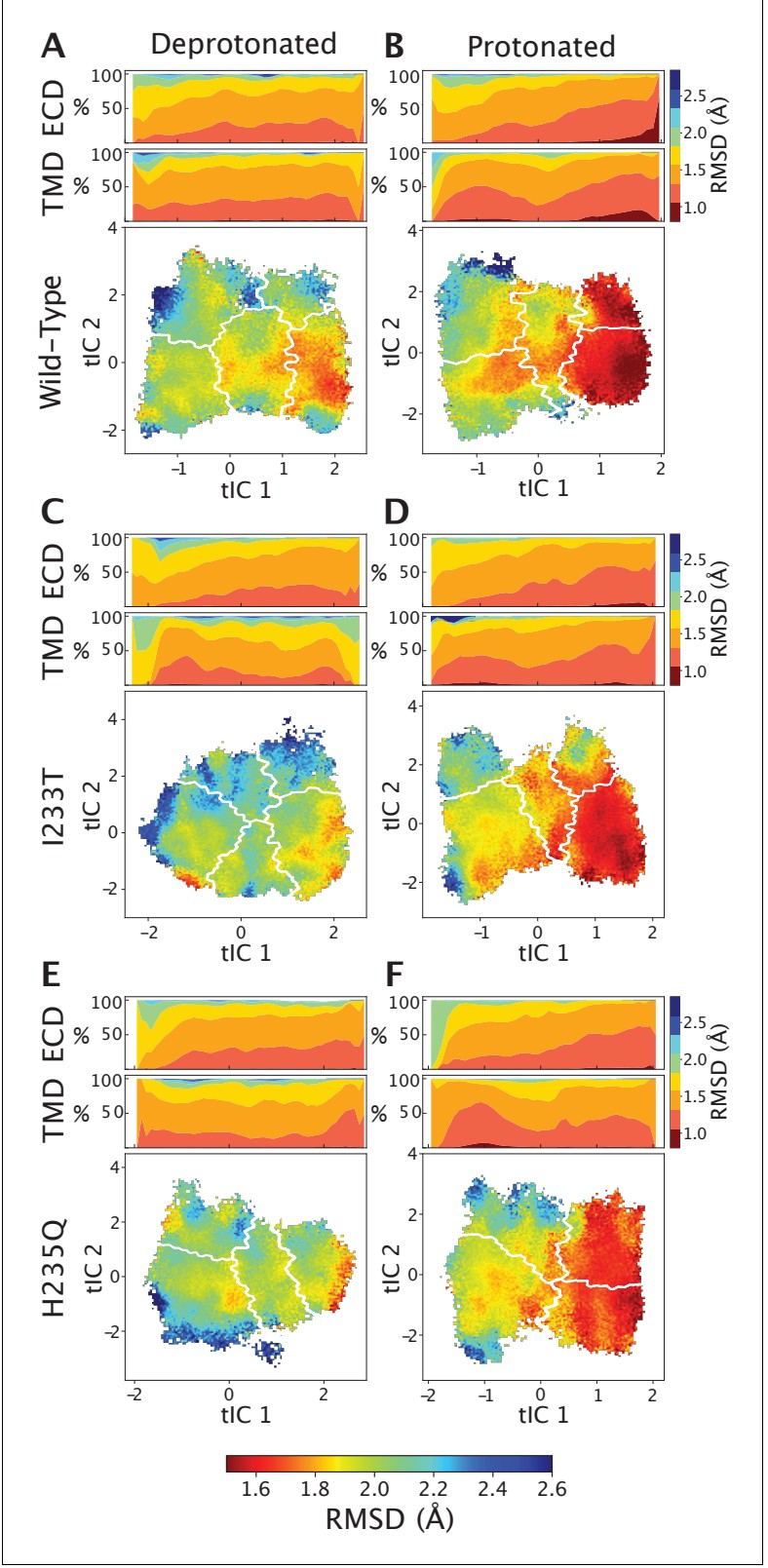

**Figure 6.** State- and protonation-dependent differences in ion channel symmetry. Heatmaps show pairwise RMSDs between all subunits of the channel, measuring the conformational symmetry of the pentamer. The two top bars show pairwise RMSDs of the transmembrane (TMD) and extracellular (ECD) domains separately, represented as stacked histograms along tIC 1. At deprotonated conditions wild-type (**A**) and the two mutations

*Figure 6 continued on next page*

*Figure 6 continued*

I233T (**C**) and H235Q (**E**) show decreased levels of symmetry in both TMD, ECD and overall. At protonated conditions wild-type (**B**), the I233T (**D**), and H235Q mutations (**F**) display high levels of symmetry in the open state coming from both ECD and TMD. Notably, the symmetry of the ECD is decreasing during channel closure and total closed state symmetry is coming mainly from the TMD.

and open X-ray structures to the SANS curve, obtained from the channel population-average in solution and at room temperature (*Lycksell et al., 2021*). This fits well with our estimate of 17%. Other channels in the pLGIC family have also been shown to attain a range of maximal open probabilities; 10–40% for GABA$_A$Rs (*Pierce et al., 2019*; *Germann et al., 2019*), 20–80% for 5-HT$_3$s (*Lambert et al., 1989*; *Mott et al., 2001*), 0.2–3% for nAChRs (*Pesti et al., 2014*; *Williams et al., 2012*), and 90–100% for GlyRs (*Ivica et al., 2021*; *Mangin et al., 2003*), when saturated with their respective natural agonist. Given the spread in open probability between pLGIC subtypes it seems reasonable that a more distant bacterial homolog like GLIC could have a unique energy landscape. Additionally, since the open probabilities for GLIC are less than 100%, protonation could function like partial rather than full agonism.

Higher order clustering enabled more detailed investigation of the different regions of the energy landscapes, including intermediate and alternative open-like and closed-like conformations. In all cases, open and modulated crystal structures projected in state IV, while closed and locally-closed states projected into state I, or on the border between state I and state II for the protonated wild-type and deprotonated H235Q datasets. Locally closed structures, characterized by an open-like ECD and closed-like TMD, projected in the same free energy basin as closed-state 4NPQ but closer to the activation free energy barrier, indicating that the locally closed state could serve as a pre-activating state in the gating pathway. Prevost et al. solved crystal structures of multiple locally closed states trapped by various mutations, and out of these only one was capable of opening in electrophysiology experiments. Additionally, a range of conformations in the M2-M3 region were sampled in the cryo-EM structures (*Prevost et al., 2012*), indicating that the locally closed conformations might not represent a separate metastable state in wild-type GLIC. Further, the H235Q mutation has been shown to crystallize in a locally closed conformation at pH 4 (*Fourati et al., 2018*). Surprisingly, our MSMs of protonated H235Q resulted in only a modest deepening of the free energy minimum around the projected locally closed structures, but also in a heightened free-energy

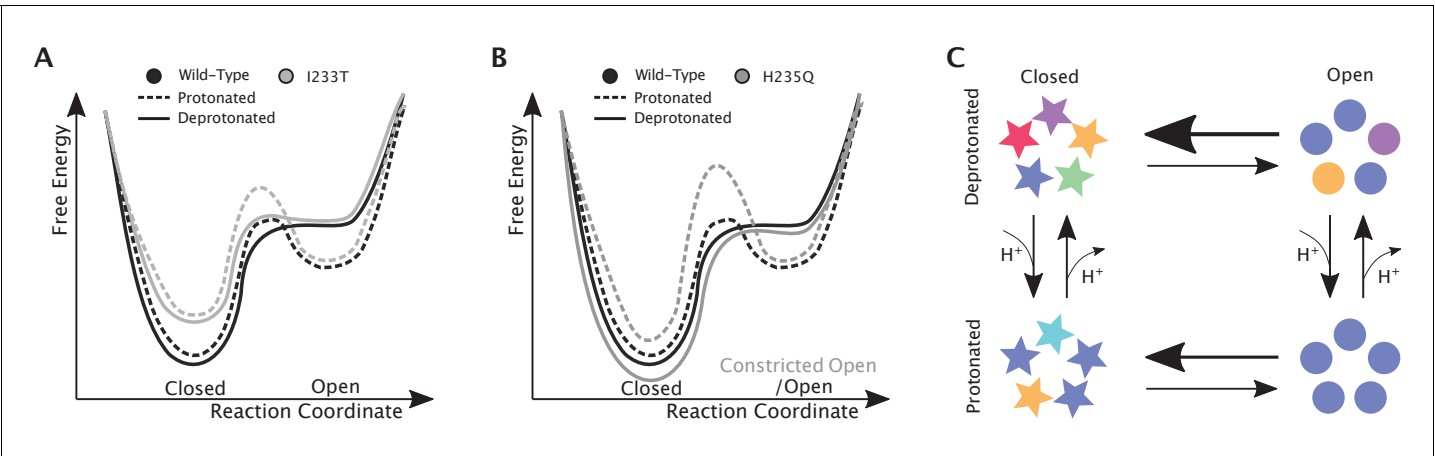

**Figure 7.** Proposed models for the free energy landscape and symmetrization in GLIC gating. Sketches of the free-energy landscapes for protonated and deprotonated (**A**) wild-type and I233T variants, and (**B**) wild-type and H235Q variants. An open-state free energy well is formed when the channel is protonated, but only a small fraction of channels will be open at any point. The I233T mutation destabilizes the closed state (**A**), while the H235Q mutation results in open channels being trapped in a state with a more constricted pore behind a higher energy barrier (**B**). (**C**) Conformational symmetries of GLIC are affected by the protonation state. Upon protonation the open state displays a high level of symmetry, which is reduced to an intermediate level in the closed state. When deprotonated, the open state achieves an intermediate level of symmetry, which is reduces to low levels of symmetry in the closed state. This suggests that protonation is important for symmetrization of the open state.

barrier between open and closed states, potentially facilitating a single state to be captured in experiments. This, in combination with our other observation that ECD compaction seems to be pH- rather than state-dependent, means that the most probable conformation for the H235Q variant at low pH has a closed-like TMD and more open-like ECD, similarly to the locally closed state.

Structural studies of the open, closed and locally closed crystal structures have revealed several conformational changes associated with GLIC gating (*Sauguet et al., 2014*). The first steps are thought to occur in the ECD through blooming and twisting motions of the entire domain. Surprisingly, our simulations did not capture state-dependent but rather pH-dependent differences in average ECD spread and twist values (*Figure 5E*; *Figure 5—figure supplement 1*; *Figure 5—figure supplement 2*) although larger fluctuations were observed in closed-like states, suggesting state-dependent differences in ECD flexibility rather than average spread and twist values. The recent closed-state cryo-EM structure solved at pH 3 (*Rovšnik et al., 2021*) displays intermediate values in ECD spread (*Figure 5E*), but a more twisted ECD compared to the closed X-ray structure (*Figure 5—figure supplement 1C*), while otherwise agreeing with values associated with the closed state TMD. While these ECD spread and twist results are somewhat counter-intuitive, the more contracted ECD could hint toward adaptation of the ECD in response to pH regardless of the TMD state. Conversely, we did observe transitions in the loops connecting the inner and outer sides of the ECD β-sandwich, which have been shown to be important for pLGIC channel function. In particular, crystal structures show breakage of the D32-R192 salt bridge in closed-state GLIC and mutational studies of D32 along with the neighboring sensor-residue E35, reveal loss of function (*Bertozzi et al., 2016*; *Nemecz et al., 2017*; *Hu et al., 2018*). In addition to capturing state-dependent differences in agreement with experimental data, our analysis also showed bimodality of the probability distributions at neutral pH for the closed states (*Figure 5D*). This effect was further enhanced upon mutation of the 9' gate, indicating that there could be an allosteric pathway between the center of the transmembrane pore and the D32-R192 salt bridge. This is also supported by previous computational models based solely on the apparent open structure (*Mowrey et al., 2013a*).

Gating motions in the TMD are particularly characterized by the tilting of the pore-lining M2 helices, leading to constriction around the 9' hydrophobic gate followed by pore dehydration. This process has previously been observed in early simulations of the TMD by *Zhu and Hummer, 2010*. Our models successfully captured how these M2 helix motions closely correlated with changes in 9' radius and pore hydration levels (*Figure 5—figure supplement 3*; *Figure 5—figure supplement 4*; *Figure 3D*). Even though we capture state-dependent differences in the 9' pore radius, and individual simulations sample pores that are as wide as the open-state structures, the most probable conformations of our open state clusters do not exhibit pores as expanded as in the majority of open crystal structures (*Figure 5B*). These structure have typically been co-crystallized with a hydrophobic plug of detergent molecules between the 9' region and the top of M2, associated with a more expanded, funnel-shaped form, which is hardly sampled in simulations after plug removal. However, I233T substitution generally increased both 9' radius to overlap better with those from the open X-ray structure, while the H235Q mutation led to overall more constriction around 9' and less hydration, potentially impacting the ability of this variant to conduct ions. Other variables that have been associated with GLIC channel gating, including kinking of the M1 helix (*Sauguet et al., 2014*) and interactions in the TMD-TMD subunit interface (*Fourati et al., 2018*), were also captured by our MSMs (*Figure 5—figure supplement 1B*; *Figure 5C*). Notably, our observations were largely consistent with (*Lev et al., 2017*), validating the use of these features as collective variables. Interestingly, the protonated alternative open-like state V displayed particular constriction at the −2' gate incompatible with ion conduction (*Figure 5—figure supplement 1A*). This is a typical feature of desensitized states in the pLGIC family (*Gielen and Corringer, 2018*), although a desensitized state for GLIC has not yet been resolved. Energetically, open and desensitized states would be expected to be separated by an energy barrier, which is not the case in our models, but it is possible state V represents conformations relevant for pre-desensitization. Since the time constant for GLIC desensitization is assumed to exceed 10 s for pH values over 3.5 (*Velisetty and Chakrapani, 2012*) and due to the lack of structures covering this state, sampling the desensitization process without applying an enhanced seeding procedure remains a difficult task.

Analysis of the conformational symmetry of GLIC revealed a particularly symmetric open state at protonated conditions (*Figure 6*). At deprotonated conditions the symmetry levels were lower overall, indicating that protonation is important for channel open-state symmetrization. In all cases,

significantly less symmetric ECDs could be observed in the closed state compared to the open state, although the difference was further enhanced at protonated conditions due to the more symmetric open state. The vast majority of GLIC structures cover the open state and display high levels of ECD symmetry, while the fewer structures covering the closed state are relatively poorly resolved, possibly indicating heterogeneity particularly in the ECD (*Sauguet et al., 2014*; *Rovšnik et al., 2021*). These results suggests that ECD symmetry could serve as an entropic driving force, where perturbation to the symmetric structure of the open state ECD facilitates channel closure. At activating conditions the ECD becomes protonated which facilitates high-level symmetrization characteristic for the open state. Furthermore, at protonated conditions the TMD displayed increased levels of symmetry both in the open state and closed state with less symmetric conformations in between. At deprotonated conditions, the symmetry level of the TMD transition was largely consistent with that of the protonated conditions, but the TMDs of the closed-like conformations were less symmetric. So far consensus has not been reached regarding whether homomeric channels transition symmetrically or asymmetrically, but evidence points toward asymmetric transitions being common. Although most structures derived from X-ray crystallography and cryo-EM are symmetric, partly due to the use of symmetric restraints during model building, there are a few examples of pLGICs with asymmetric structures. Recent cryo-EM structures of the 5-HT$_{3A}$ receptor in lipid nanodiscs revealed an asymmetric closed state, whereas a symmetrized open state was stabilized by serotonin molecules bound at all five ECD sites (*Zhang et al., 2021*). Additionally, a cryo-EM structure of the resting state *Torpedo* nAChR with toxins bound to two of the five subunits displayed asymmetries in both the ECD and TMD (*Rahman et al., 2020*), and cryo-EM structures of the GABA$_A$ receptor displayed ECD asymmetry in the resting state compared to the desensitized where ligands were bound (*Masiulis et al., 2019*). When it comes to computational work, long (15–20 μs) simulations of 5-HT$_{3A}$ have shown evidence of asymmetric closure of the TMD pore upon channel pre-activation (*Guros et al., 2020*). Mowrey et al. also proposed that asymmetric propofol binding could create unbalanced forces such that symmetry breaking would facilitate channel conformational transitions (*Mowrey et al., 2013b*). In our simulations, ligands (protons) were bound symmetrically across all subunits which further indicates that asymmetric transitions could be important regardless of symmetric or asymmetric ligand-binding. Understanding these asymmetries is nonetheless important for understanding the gating mechanism, but could also bring clarity regarding differences between homomeric and heteromeric pLGICs, as well as effects of different permutations of subunit assembly.

One limitation of our simulations was that protonation states were fixed throughout the simulations and selected according to consensus from previous studies. Indeed, protonation of some of titratable residues are expected to have a large effect on the results, while others are expected to be less important. Mutational studies of all titratable residues in GLIC identified that mutation of E26, D32, E35, D122, E222, H235, E243, and H277 altered GLIC activation, and out of these E35 was identified as the main proton sensor (*Nemecz et al., 2017*; *Hu et al., 2018*). In our simulations, protonation of E35 indeed had a large effect on the local backbone conformation, suggesting importance for proton-sensing (*Figure 5F*). Previous $pK_a$ calculations generated the following most probably protonation pattern at pH 4.6; E26, E35, E67, E75, E82, D86, D88, E177, E243, and H277 doubly protonated (*Nury et al., 2011*), which is also the pattern used in this paper. Other simulation studies have used similar patterns with slight modifications to model low pH conditions; E69 protonated (*Nury et al., 2010*), E69 protonated and three or four among five randomly chosen symmetrical copies of D/E residues protonated (*Cheng et al., 2010*), H127 doubly protonated (*Prevost et al., 2012*), E69 protonated and H127 doubly protonated (*Calimet et al., 2013*), E69 protonated, H127 doubly protonated, and D86 and D88 deprotonated (*Lev et al., 2017*), and D13, D31, D55, D91, D97, D145, D153, D154, D161, D178, D185, E14, E69, E147, E163, E181, E272, and H127 doubly protonated (*Fritsch et al., 2011*). Although these simulations were primarily covering the open state, we see no large discrepancies in results and can therefore expect our simulations to be generally insensitive to the permutations in protonated residues mentioned above. Once the method has matured, constant pH simulations will likely solve the combinatorial problem that protonation state selection creates (*Chen et al., 2014*).

In summary, our Markov states models are able to predict shifts in free energies upon change in activating stimulus (pH) as well as gain-of-function and loss-of-function mutations. The models predict relatively low values of maximal open probability, with relative differences in agreement with

electrophysiology recordings. Our simulations also captured state-dependent differences in previously proposed mechanistic variables as well as state- and pH-dependent differences in channel conformational symmetry. Due to the possibility of direct comparison and validation to electrophysiology, these models are able to predict high-level statistical properties together with the details of channel conformational changes. This enables further exploration of changes that correlate with gating to further the fundamental understanding of pLGIC gating and, potentially, the development of state-selective drugs. We expect this methodology to be transferable to other channels with potential to predict properties from electrophysiology, even in absence of full equilibrium sampling.

## Materials and methods

### Pathway construction using eBDIMS

To enhance sampling, 50 seeds along the GLIC closed–open gating pathway were obtained along forward and reverse elastic-network driven Brownian dynamics (eBDIMS) simulations (*Orellana et al., 2016*; *Orellana et al., 2019*). Here, the channel was represented as an elastic network model using the Cα representation of apparent closed (PDB ID 4NPQ) (*Sauguet et al., 2014*) and open (PDB ID 4HFI) (*Sauguet et al., 2013*) structures, and driving transitions in both directions by progressively minimizing the difference in internal distances between the current and the target states. Langevin dynamics with implicit solvent and harmonic forces was used to model the dynamics, with a 12 Å intra-subunit and 8 Å inter-subunit cutoff distance, respectively. The force constant of the elastic network was 10 kcal/(mol·Å$^2$), as previously used in *Orellana et al., 2016*.

### Model reconstruction

The atomistic detail of the seeds was reconstructed using Modeller version 9.22 (*Sali and Blundell, 1993*) in two steps. First, side chain atoms from the template X-ray structure (PDB ID 4HFI) were added to each model, followed by a cycle of refinement with all Cα atoms restrained. Restraints on Cα atoms were then substituted with restraints on backbone hydrogen bonds, taken from helix and sheet annotations in the template PDB file, for another cycle of refinement to ensure proper secondary structure.

### Molecular dynamics simulations

The reconstructed seed models were embedded in a palmitoyloleoylphosphatidylcholine (POPC) lipid bilayer, solvated by water and 0.1 M NaCl. Activating conditions were modeled by protonation of a subset of acidic residues (E26, E35, E67, E75, E82, D86, D88, E177, E243; H277 doubly protonated) to approximate the probable pattern at pH 4.6, as previously described (*Nury et al., 2011*). All systems were energy minimized with steepest descent for 10,000 steps. NPT equilibration was carried out in four steps, restraining all heavy atoms in the first cycle, then only backbone atoms, then Cα atoms, and finally the M2 helices, for a total of 76 ns. Equilibration and production runs were performed using the Amber99SB-ILDN force field with Berger lipids, together with the TIP3P water model. Temperature coupling was achieved with velocity rescaling (*Bussi et al., 2007*) and pressure coupling with the Parrinello-Rahman barostat (*Parrinello and Rahman, 1981*). The simulations were prepared and run with GROMACS versions 2018.4 and 2019.3 (*Abraham et al., 2015*) for 1.2 µs each, allowing for each individual trajectory to substantially deviate from the starting seed conformation and collectively sample space around the transition pathway broadly (*Figure 2—figure supplement 1*).

### Markov state models

The ion channel was described with a set of 1585 features, including interatomic distances within and between subunits. Since GLIC is a symmetric homopentamer, we introduced a feature transformation where all distances originating from one subunit were scored as an independent trajectory; thus each trajectory contained information as to how one subunit moved in relation to all others. Dimensionality reduction was achieved using tIC analysis, previously shown to form a good basis for discretization of conformational space into Markov states, as the tIC eigenvetors linearly approximate the MSM eigenvectors (; *Schwantes and Pande, 2013*). Typically, MSMs are constructed with

the assumption of exhaustive sampling of the equilibrium distribution, and inclusion of as much kinetic information as possible. Although suitable for peptide-sized systems, we found it practically unfeasible for the large-scale motions in ion channels since the MSM tends to optimize for slow but undersampled processes that may not be of primary interest. Accordingly, we discretized the state space into a few, but meaningful, tIC dimensions by omitting faster dimensions where the data was represented as a single Gaussian. Hyperparameter optimization is commonly solved by maximizing the variational approach for Markov porcesses (VAMP) (*Wu and Noé, 2019*) score with cross-valida-tion to avoid overfitting; however, in our case, we found that VAMP favored exploration of slower and undersampled processes rather than convergence of a few timescales of interest, in the absence of of exhaustive equilibrium sampling. Instead, we relied on a simpler elbow approach to select appropriate hyperparameters, by calculating the open probabilities using PCCA+ (*Röblitz and Weber, 2013*) for various hyperparameter combinations. Based on the resulting plots (*Figure 2—figure supplement 3A*), we selected 300 clusters for kMeans clustering from the 'elbow' of the plot to ensure convergence and avoid overfitting. The tIC lag times yielded consistent results within a 5–25 ns range for all datasets (*Figure 2—figure supplement 3B*), so a 20 ns lag time was selected, with kinetic mapping (*Noé and Clementi, 2015*). Instead of variationally optimizing all eigenvalues (as in VAMP), we plotted the slowest implied timescale of greatest interest using different hyper-parameter combinations (*Figure 2—figure supplement 3C*). Indeed, our previous selection was optimal for the deprotonated conditions, so we used these parameters for all six datasets for consis-tency. We also note that the timescales of the selected model and the variationally optimal one were almost within the error margin for the protonated conditions. The MSM lag time was deter-mined to 20 ns from the implied timescales, by selecting the shortest possible Markovian lag time to maximize resolution of the MSM processes (*Figure 2—figure supplement 2*). Two- and four-or-five-state macrostate models were obtained through PCCA+ clustering of the MSM eigenvectors (*Röblitz and Weber, 2013*). For the two-state model, the number of macrostates was selected in accordance to the number of observed energy minima (*Figure 2*) so that macrostates would repre-sent metastable states. Higher-order clustering was done to enable more fine-grained analysis of conformations at more kinetically different regions of the phase space, and particularly to enable extraction of an intermediate cluster (*Figure 4*, State III). The choice of number of macrostates for these models were dependent on the structure of the MSM eigenvectors, and optimized to enable comparability between datasets and avoid macrostates with too small populations. One simulation from the protonated wild-type dataset was rejected from analysis due to obstruction of the tIC land-scape with a slower but undersampled process, leaving 58.8 μs total sampling; MSMs of the other three conditions (deprotonated wild-type, deprotonated and protonated I233T, deprotonated and protonated H235Q) contained 60 μs sampling each. Sampling and convergence of each MSM were assessed through convergence of the slowest implied timescales (*Figure 2—figure supplement 2A, D*), Chapman-Kolmogorov tests (*Prinz et al., 2011*; *Figure 2—figure supplement 2C,F*) and by assessing the level of reversible sampling achieved (*Figure 2—figure supplement 2B,E*). Markov state modeling was done with PyEMMA version 2.5.7 (*Scherer et al., 2015*).

## Electrophysiology

Two-electrode voltage-clamp electrophysiology was performed as previously described (*Heusser et al., 2018*). Briefly, GLIC cDNA subcloned in vector pMT3 was modified using commer-cially synthesized primers (Invitrogen, Stockholm, Sweden) and the GeneArt site-directed mutagene-sis system (Thermo Fischer, Waltham, MA). Plasmids were amplified using a HiSpeed Plasmid Purfication Midi kit (Qiagen, Hilden, Germany), and verified by cycle sequencing (Eurofins Genomics GmbH, Ebersberg, Germany). Nuclei of isolated stage V–VI *Xenopus laevis* oocytes (EcoCyte BioSci-ence, Dortmund, DE) were injected with 0.5–3.0 ng cDNA and stored in incubation medium (88 mM NaCl, 10 mM HEPES, 2.4 mM NaHCO3, 1 mM KCl, 0.91 mM CaCl2, 0.82 mM MgSO4, 0.33 mM Ca (NO3)2, 2 mM sodium pyruvate, 0.5 mM theophylline, 0.1 mM gentamicin, 17 mM streptomycin, 10,000 u/L penicillin, pH 8.5) for 2–7 days. Glass electrodes were pulled and filled with 3 M KCl to reach an initial resistance of 0.1–0.5 MΩ. Expressing oocytes were clamped at −70 mV using an OC-725C voltage clamp (Warner Instruments, Hamden, CT, USA), and perfused with running buffer (123 mM NaCl, 10 mM HEPES, 2 mM KCl, 2 mM MgSO$_4$, 2 mM CaCl2, pH 8.5) at a flow rate of 0.35 mL/min. Activation buffers contained 10 mM MOPS or citrate in place of HEPES, adjusted in 0.5 pH unit increments, and were exchanged for running buffer via a VC$^3$-8 valve controlled pressurized

perfusion system (ALA Scientific Instruments, Farmingdale, NY, USA). Currents were digitized at a sampling rate of 5 kHz with an Axon CNS 1440A Digidata system using pCLAMP 10 (Molecular Devices, Sunnyvale, CA, USA). Each pH response was measured as the peak current after 1 min exposure to activation buffer, normalized to the same oocyte's response to the lowest pH tested. Each reported value represents the mean from six to eight oocytes, ± standard error of the mean. Proton concentration-dependence curves were fit by nonlinear regression with bottom and top restraints using Prism 8.0 (GraphPad Software, La Jolla, CA). Measurements from variant H235Q, collected by equivalent methods, are reproduced here from our previously published data (*Fourati et al., 2018*).

## Acknowledgements

This work was supported by grants from the Knut and Alice Wallenberg Foundation, the Swedish Research Council (2017–04641, 2018–06479, 2019–02433), the Swedish e-Science Research Centre, and the BioExcel Center of Excellence (EU 823830). Computational resources were provided by the Swedish National Infrastructure for Computing. The authors are grateful to Joseph Jordan, Brooke Husic, and Lucie Delemotte for helpful feedback and discussions.

## Additional information

### Funding

| Funder | Grant reference number | Author |
| --- | --- | --- |
| Knut och Alice Wallenbergs Stiftelse | | Erik Lindahl |
| Vetenskapsrådet | 2017-04641 | Erik Lindahl |
| Vetenskapsrådet | 2018-06479 | Erik Lindahl |
| Vetenskapsrådet | 2019-02433 | Erik Lindahl |
| Swedish Research Council | | Rebecca Howard Erik Lindahl |
| Horizon 2020 | BioExcel (823830) | Erik Lindahl |
| Swedish National Infrastructure for Computing | 2020/3-37 | Erik Lindahl |

The funders had no role in study design, data collection and interpretation, or the decision to submit the work for publication.

### Author contributions

Cathrine Bergh, Data curation, Software, Formal analysis, Validation, Investigation, Visualization, Methodology, Writing - original draft, Writing - review and editing; Stephanie A Heusser, Data curation, Formal analysis, Validation, Investigation, Visualization, Methodology, Writing - review and editing; Rebecca Howard, Conceptualization, Data curation, Formal analysis, Supervision, Validation, Investigation, Visualization, Methodology, Writing - original draft, Project administration, Writing - review and editing; Erik Lindahl, Conceptualization, Resources, Software, Formal analysis, Supervision, Funding acquisition, Methodology, Writing - original draft, Project administration, Writing - review and editing

### Author ORCIDs

Cathrine Bergh https://orcid.org/0000-0001-7540-5887
Stephanie A Heusser https://orcid.org/0000-0003-3224-4547
Erik Lindahl https://orcid.org/0000-0002-2734-2794

### Decision letter and Author response

Decision letter https://doi.org/10.7554/eLife.68369.sa1
Author response https://doi.org/10.7554/eLife.68369.sa2

## Additional files

### Supplementary files
- Transparent reporting form

### Data availability

Additional data including simulation parameters, Markov state models, sampled conformations and full trajectories can be accessed at https://doi.org/10.5281/zenodo.5500174.

The following dataset was generated:

| Author(s) | Year | Dataset title | Dataset URL | Database and Identifier |
|---|---|---|---|---|
| Bergh C, Heusser SA, Howard R, Lindahl E | 2021 | Markov State Models of Proton- and Pore-Dependent Activation in a Pentameric Ligand-Gated Ion Channel | https://doi.org/10.5281/zenodo.5500174 | Zenodo, 10.5281/zenodo.5500174 |

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
