## [Decision Letter]

**Acceptance summary:**

This article presents state-of-the-art molecular dynamics simulations of the pH-gated pentameric ion channel GLIC, which has been the subject of many structural and functional studies. GLIC can be considered as a model system for pentameric ligand-gated ion channels that are responsible for fast chemical-electrical communication between cells in animals. The findings include the solution of open- and closed-like channel forms, intermediates and a "pre-desensitised" state. The approach reproduces modulation by pH and opposing mutations, correctly reproducing loss or gain of function, representing convincing proof of the success of the approach. Overall, the sampling of channel dynamics is significant and the description of state interconversions sheds new light on pLGIC mechanisms. The reviewers were convinced by the substantial changes and additions to the manuscript, as well as the new insight into the roles of pH and mutations in GLIC function. Concerns over convergence, sampling and methods descriptions have been allayed and the manuscript is now suitable for publication.

**Decision letter after peer review:**

Thank you for submitting your article "Markov State Models of Proton-and Gate-Dependent Activation in a Pentameric Ligand-Gated Ion Channel" for consideration by *eLife*. Your article has been reviewed by 3 peer reviewers, including Toby Allen as Reviewing Editor and Reviewer #1, and the evaluation has been overseen by José Faraldo-Gómez as the Senior Editor. The following individual involved in review of your submission has agreed to reveal their identity: Frédéric Poitevin (Reviewer #3).

Essential revisions:

While the reviewers agree on the strengths and importance of the findings, the following changes or additions are required:

1. Analyse one the numerous mutants that are blocked in the so-called locally-closed state, thereby helping the community to understand the nature of this often-encountered phenotype.

2. Demonstrate convergence and sampling, including proof of lack of dependence on the initial path.

3. Better describe of the nature of the observed states (especially the locally closed state) and the tICA gating coordinates.

4. Additional methods description and clarification is needed, with clearer relation to past simulation studies to avoid misapprehensions.

*Reviewer #1 (Recommendations for the authors):*

The authors state that the problem with some past methods is a presumed pathway (which should be distinguished from a presumed finite set of coordinates/collective variables, as in Ref.20). Yet the MSM is not without any assumptions. The seeding of simulations in any MSM, including here, stems from an initial presumed gating transition from some simplified model. See above comments.

In the beginning of the Results section (page 5) we see a typical representation of free energy within a tICA1/2 space. See above comment. Can the key tICA1 vector be visualised, say with a vector mapping (movements of residues…) on the structure to allow it to be more readily interpreted? Moreover, it is not clear what tIC2 represents by analysis of figure 2 and later figures, and its importance to gating not immediately obvious. Is there a physical interpretation for it? Overall, I find the discussion around figure 2 to be wanting of more physical interpretation.

The authors find predominantly closed channels, although I233T (which shifts pH50 to allow conduction even at neutral pH in experiments) revealed a second open-like free energy minimum. Protonation revealed a much broader second open minimum (though still by far predominantly closed, even at low pH). This predominant closure was not seen in in past string solutions of Lev et al., and it is not clear how this is consistent with the observed pH50 for wildtype GLIC experimentally. See above comment on this, relating to the preprint in Ref.33, which appears to be the main evidence used by the authors for a low probability of a conducting form.

Regarding other states, I feel the intermediate states (such as those of states I-V with partial TMD/ECD change) have not been well characterised, or their implications explained as well as could be. Also, the authors could better visualise the location of any LC forms on the maps and explain their roles in gating (see above comments). In figure 2 the LC states are said to lump in with closed state structures (page 6). The dots in figure 4 have no colours. I guess some of the left dots are LC? See my comments above about confusion over LC. It is particularly interesting that one state with -2' constriction may relate to the desensitised state (although the authors see no barrier towards it, and so its relation to a desensitised state is not clear). Why the method cannot sample the actual desensitised state is not well explained – though may suggest a limitation in the analysis based on microsecond trajectories.

Analysis of states against past variables (as in Lev et al., Ref 20) mostly goes as expected, although the ECD behaviour does not (page 10). Though pH dependence is seen, the MSM does not reveal clear bloom and twist diffs O-C states and the authors suggest it is an artefact of x-ray crystals. While twist was less obvious, ECD spreading did differ between states in the past model based on strings by Lev et al., Is this because the finite space of Lev et al., or becoming trapped near those values? Why did the ECD spread effect disappear in this current model? Was it present after the initial seeding but vanished in subsequent MD libraries, or did this come from the initial seeding procedure?

Symmetry of the ECD decreases in closed state, and at high pH. This is as to be expected based on the higher structural diversity (and symmetry loss) of the ECD seen in Xray of the closed/high pH structure. A lot has been made about the role of asymmetry. Can the authors better explain to the reader why asymmetry in gating really matters?

In comparison to the previous string method on page 11, the authors write that barriers like 1-1.5 and 1.5-2 are lower than 1.5 and 2.5 kcal/mol, suggesting the current model extracted lower barriers – presumably being better sampled. However, given the errors I would just say they are consistent.

The statement on page 4 that Lev et al., only used short picosecond timescale simulations in their string method could be misleading for the readers. The string method used by Lev et al., makes use of large libraries of 5 and 20ps trajectories to sample the directions of change, but are carried out in parallel and repeated for many hundreds of iterations, such that the timescale sampled is much longer than that suggested by the authors here. More critically, the string method is an enhanced sampling approach such that timescales of motions cannot be judged by these values, as they adaptively sense the directions of change, allowing the approach to explore configurational space well beyond the actual simulation times. In fact, one might suggest the MSM approach used here, based on libraries of 1 microsecond free trajectories, may be limited in the states they can explore for processes that occur on much longer timescales. While it is always possible to have more sampling of random trajectories in the string method, I would say the main advantage of the present MSM is the lack of presumption of order parameters, which could be further explored in string methods. I do note, however, that one of the conclusions here (on page 12) is that the variables used by Ref.20 were consistent with the findings here.

Regarding the methods, overall the techniques used could be better explained, both for the MD/MSM expert and the general reader, avoiding jargon and relying on packaged methods, and better justifying the choices made. Things like eBDIMS, TIC etc will not mean much to many readers in the current form. The motivations for the choices and their details are important to this study.

*Reviewer #2 (Recommendations for the authors):*

The authors should take the opportunity of their working methodology to explain to the community the role of the LC state: is it an on-pathway form in which some of the mutants get stuck? or is it something different (off-pathway)?

– To this effect, an in-depth study of one of the many known LC mutants is needed.

– p. 2 There is a rather bold and questionable general statement in p. 2 about experimental structures, stating that "the stability required for crystal packing or cryo-EM data processing results in structures mostly representing metastable states".

I strongly suggest replacing the word "mostly" by "sometimes".

*Reviewer #3 (Recommendations for the authors):*

My main recommendations to the authors are very general and do not require to be answered for acceptance of publication.

Regarding the apparent subjectivity in the clustering choices and the difficulty/impossibility of exhaustive sampling for this kind of system, it would be interesting to motivate the choices made (5 states, PCCA+, …). For example, would the authors recommend their approach for simulations involving other large systems – can it be transferred – or was it thought in reaction to the particular sampling obtained here? Elements of answers to this question have the potential to help overcome the problem of robust analysis of undersampled datasets, which could benefit greatly the simulation community.

It is somewhat unclear to me what the new insights exactly are:

– The idea that protonation is mainly a driver of compaction, not of the gating is interesting. Unfortunately, there is no discussion of the combinatorial problem that protonation poses or mention of constant pH simulations that would be more realistic than applying fixed protonation that were proposed in the past – exploring the effect of a few selected protonation state changes would be interesting to see if the results are robust to a few perturbations.

– Discussion about desensitized state is interesting; it would have been nice to hear more from the authors about their take on the possibility to access this regime with simulations. In principle MSM have the potential to yield kinetic models that could be directly compared to the ones extracted from electrophysiology; what is the authors take on this? Will that ever be possible? When? Would a more experimental approach be desirable, instead of post-hoc comparison?

---

## [Author Response]

Essential revisions:While the reviewers agree on the strengths and importance of the findings, the following changes or additions are required:1. Analyse one the numerous mutants that are blocked in the so-called locally-closed state, thereby helping the community to understand the nature of this often-encountered phenotype.

As we communicated earlier, this was indeed a very worthwhile effort despite the efforts required. We have completed an additional 120 µs simulations of the H235Q mutant known to crystallize in a locally closed state at low pH. We constructed two additional MSMs of this variant at deprotonated and protonated conditions and analyzed the models in a similar fashion as our four previous MSMs. We have updated Figures 1-7, Figure 2—figure supplement 1-4, Figure 3—figure supplement 3, Figure 5—figure supplement 1, and Figure 5—figure supplement 3-11 with data related to this variant. We have also extended the results and Discussion sections to include these new results. While some caveats apply (the tIC coordinates we use primarily describe the normal gating, and they are not *identical* for different mutants), there are also clear differences related to the local minima vs. transition barrier and hydration that we discuss in the text. Thank you for suggesting this; it turned out to be a very nice extension for the work!

2. Demonstrate convergence and sampling, including proof of lack of dependence on the initial path.

Good point. We want to be a little bit careful about absolute claims, since no method is able to truly reach convergence or be completely independent of starting conditions for systems of this size, but we have added three new figures that enable the reader to assess both sampling and how far away from the initial path we sample: Figure 2—figure supplements 1-3 together with relevant discussion in the methods section.

3. Better describe of the nature of the observed states (especially the locally closed state) and the tICA gating coordinates.

We have added more descriptions of conformations not represented by experimental structures in the results and Discussion sections. Additionally, we have added a paragraph discussing the role of the locally closed state. Regarding the tICA coordinates we added Figure 2—figure supplement 4 to clarify the meaning of the tICA coordinates.

4. Additional methods description and clarification is needed, with clearer relation to past simulation studies to avoid misapprehensions.

Very reasonable point; there are a number of groups that deserve credit in this field! We have substantially rewritten the methods section to better describe our methodological choices. We have also added three additional references to GLIC simulation studies and better clarified their similarities or differences in the introduction and Discussion sections.

Reviewer #1 (Recommendations for the authors):The authors state that the problem with some past methods is a presumed pathway (which should be distinguished from a presumed finite set of coordinates/collective variables, as in Ref.20). Yet the MSM is not without any assumptions. The seeding of simulations in any MSM, including here, stems from an initial presumed gating transition from some simplified model. See above comments.

Very fair point; we were too brief when justifying our choice of MSMs. While they have a (neat) ability to move pathways across local maxima, and allowing multiple paths, this is of course very dependent on sampling, which we hope is now better covered with our updated figures and discussion, as well as the introduction covering more of earlier studies and alternative approaches.

In the beginning of the Results section (page 5) we see a typical representation of free energy within a tICA1/2 space. See above comment. Can the key tICA1 vector be visualised, say with a vector mapping (movements of residues…) on the structure to allow it to be more readily interpreted? Moreover, it is not clear what tIC2 represents by analysis of figure 2 and later figures, and its importance to gating not immediately obvious. Is there a physical interpretation for it? Overall, I find the discussion around figure 2 to be wanting of more physical interpretation.

We have followed the reviewer’s suggestion and projected vector mappings onto the structure in Figure 2—figure supplement 4.

The authors find predominantly closed channels, although I233T (which shifts pH50 to allow conduction even at neutral pH in experiments) revealed a second open-like free energy minimum. Protonation revealed a much broader second open minimum (though still by far predominantly closed, even at low pH). This predominant closure was not seen in in past string solutions of Lev et al., and it is not clear how this is consistent with the observed pH50 for wildtype GLIC experimentally. See above comment on this, relating to the preprint in Ref.33, which appears to be the main evidence used by the authors for a low probability of a conducting form.

Thanks for raising this important point! It is certainly interesting that the string method and MSMs gave different results related to open probabilities. Although our electrophysiology recordings are not able to reveal the fraction of open channels at maximal conduction, the shifts in open probabilities seen in our MSMs *are* consistent with all shifts observed in the electrophysiology recordings, also for the two additional MSMs of the loss-of-function mutation added in this revision. Additionally, these results are also in line with a new SANS study from our lab with an experimental estimation of the open probability (ref. 44) that was published after our initial submission. We have now added this reference to our discussion:

“Additionally, recent small-angle neutron scatting (SANS) experiments estimated low-pH P_open_ to 18% by fitting linear combinations of closed and open X-ray structures to the SANS curve, obtained from the channel population-average in solution and at room temperature [44]. This fits well with our estimate of 17%.”

Regarding other states, I feel the intermediate states (such as those of states I-V with partial TMD/ECD change) have not been well characterised, or their implications explained as well as could be.

We agree that the features of intermediate conformations could be more extensively described. First, we have described more features of state III in text, particularly on pages 6-9 and 12. Regarding the locally closed state, we have extended the results and discussion with new simulations of the H235Q mutation to better address the questions of ECD spread in relation to the locally closed state.

Also, the authors could better visualise the location of any LC forms on the maps and explain their roles in gating (see above comments). In figure 2 the LC states are said to lump in with closed state structures (page 6). The dots in figure 4 have no colours. I guess some of the left dots are LC? See my comments above about confusion over LC.

We again regret that our figures were unclear. We have therefore updated Figure 2, Figure 3 and Figure 4. We have also extended the discussion about the locally closed state.

It is particularly interesting that one state with -2' constriction may relate to the desensitised state (although the authors see no barrier towards it, and so its relation to a desensitised state is not clear). Why the method cannot sample the actual desensitised state is not well explained – though may suggest a limitation in the analysis based on microsecond trajectories.

We too are definitely excited about this, while some caution should be applied! We have added a sentence to the discussion detailing the unfeasibility of sampling the desensitization process. We also wish to clarify that the limitation lies in sampling and not analysis:

“Since the time constant for GLIC desensitization is assumed to exceed 10s for pH values over 3.5 [17] and due to the lack of structures covering this state, sampling the desensitization process without applying an enhanced seeding procedure remains a difficult task.”

Analysis of states against past variables (as in Lev et al., Ref 20) mostly goes as expected, although the ECD behaviour does not (page 10). Though pH dependence is seen, the MSM does not reveal clear bloom and twist diffs O-C states and the authors suggest it is an artefact of x-ray crystals. While twist was less obvious, ECD spreading did differ between states in the past model based on strings by Lev et al. Is this because the finite space of Lev et al., or becoming trapped near those values? Why did the ECD spread effect disappear in this current model? Was it present after the initial seeding but vanished in subsequent MD libraries, or did this come from the initial seeding procedure?

The point about ECD behavior is an important one and we regret this was not clearer in our initial submission. We also wish to clarify that we do not think differences in ECD spread and twist are artifacts in X-ray crystals: we do see differences, but not as extreme as in most structures. From our MSMs we see pH-dependent rather than state-dependent differences in ECD spread and twist values, suggesting that changes in these variables are a result of pH-sensing. To support this argument, we added Figure 5-supplement 2 with time-series of the ECD spread and twist values for all seeded wild-type simulations with starting ECD spread and twist values interspersed between those for 4HFI and 4NPQ. From this figure we see rapid adaptation (50 ns) of the ECD spread and twist values to averages distinct for each pH value. However, we *do* see a difference in the fluctuations of these variables based on the starting conformation, but the average is generally not affected.

Regarding comparison to the results from Lev et al., ECD spread and twist differences did not disappear from the MSMs, but represent a fast motion as the system equilibrates to the pH value of the stationary distribution. Since the free energy landscapes from our MSMs represent the stationary distribution at a certain pH value after long times, the ECD spread and twist do not show up as part of the gating mechanism in the stationary distribution, but rather as a prerequisite to reach that stationary distribution, as indicated by pH-dependent shifts in averages between protonated and deprotonated datasets. Since Lev et al., used collective variables rather than kinetic models, the fast equilibrating motions of the ECD spread might very well show up in the beginning of the gating mechanism. We extended the discussion on page 12 to include more detail about this.

Symmetry of the ECD decreases in closed state, and at high pH. This is as to be expected based on the higher structural diversity (and symmetry loss) of the ECD seen in Xray of the closed/high pH structure. A lot has been made about the role of asymmetry. Can the authors better explain to the reader why asymmetry in gating really matters?

We have followed the advice of the reviewer and added the following sentence to the discussion:

“Understanding these asymmetries is nonetheless important for understanding the gating mechanism, but could also bring clarity regarding differences between homomeric and heteromeric pLGICs, as well as effects of different permutations of subunit assembly.”

In comparison to the previous string method on page 11, the authors write that barriers like 1-1.5 and 1.5-2 are lower than 1.5 and 2.5 kcal/mol, suggesting the current model extracted lower barriers – presumably being better sampled. However, given the errors I would just say they are consistent.

We definitely agree with this (although 1kcal/mol is a 5-fold difference in kinetics!) and now refer to the free energy barriers as “comparable” to those of Lev et al., rather than “slightly lower”:

“At neutral pH the barrier from string method hydration gave 2.5 kcal/mol, while the MSM resulted in 1.5-2.0 kcal/mol height of the energy barrier. This indicated that the wild-type MSMs found transition pathways with comparable free energy barriers.”

The statement on page 4 that Lev et al., only used short picosecond timescale simulations in their string method could be misleading for the readers. The string method used by Lev et al., makes use of large libraries of 5 and 20ps trajectories to sample the directions of change, but are carried out in parallel and repeated for many hundreds of iterations, such that the timescale sampled is much longer than that suggested by the authors here. More critically, the string method is an enhanced sampling approach such that timescales of motions cannot be judged by these values, as they adaptively sense the directions of change, allowing the approach to explore configurational space well beyond the actual simulation times. In fact, one might suggest the MSM approach used here, based on libraries of 1 microsecond free trajectories, may be limited in the states they can explore for processes that occur on much longer timescales. While it is always possible to have more sampling of random trajectories in the string method, I would say the main advantage of the present MSM is the lack of presumption of order parameters, which could be further explored in string methods. I do note, however, that one of the conclusions here (on page 12) is that the variables used by Ref.20 were consistent with the findings here.

Sorry; this was not meant as a comment about the relative merits of the methods, but rather an attempt at very briefly summarize approaches. Indeed, just as MSMs enable sampling of kinetic processes far beyond the timescales of individual trajectories, so does the string method! We have removed “short ps-timescale” so that the sentence in the introduction now reads:

“This provides precious information of structural rearrangements, but the choice of collective variables in combination with the timescales of individual simulations may influence what motions are sampled.”

Regarding the methods, overall the techniques used could be better explained, both for the MD/MSM expert and the general reader, avoiding jargon and relying on packaged methods, and better justifying the choices made. Things like eBDIMS, TIC etc will not mean much to many readers in the current form. The motivations for the choices and their details are important to this study.

Check. We have added explanations to terms like tICA and eBDIMS in the beginning of the Results section and rewritten the methods section to better motivate choices.

Reviewer #2 (Recommendations for the authors):The authors should take the opportunity of their working methodology to explain to the community the role of the LC state: is it an on-pathway form in which some of the mutants get stuck? or is it something different (off-pathway)?– To this effect, an in-depth study of one of the many known LC mutants is needed.

In hindsight, we appreciate being pushed to do this! We have run additional simulations of the H235Q mutant. Regarding its relation to the LC state, we added a paragraph to the discussion on page 12.

– p. 2 There is a rather bold and questionable general statement in p. 2 about experimental structures, stating that "the stability required for crystal packing or cryo-EM data processing results in structures mostly representing metastable states".I strongly suggest replacing the word "mostly" by "sometimes".

We have followed the suggestion from the reviewer so that the sentence now reads:

“since the stability required for crystal packing or cryo-EM data processing results in structures sometimes representing metastable states.”

Reviewer #3 (Recommendations for the authors):My main recommendations to the authors are very general and do not require to be answered for acceptance of publication.Regarding the apparent subjectivity in the clustering choices and the difficulty/impossibility of exhaustive sampling for this kind of system, it would be interesting to motivate the choices made (5 states, PCCA+, …). For example, would the authors recommend their approach for simulations involving other large systems – can it be transferred – or was it thought in reaction to the particular sampling obtained here? Elements of answers to this question have the potential to help overcome the problem of robust analysis of undersampled datasets, which could benefit greatly the simulation community.

Good idea. We have expanded the motivations of various choices in the methods section and particularly included motivations for the number of PCCA+ clusters used in our analyses. We also added a sentence with our opinion on the potential transferability of the methodology to similar systems:

“We expect this methodology to be transferable to other channels with potential to predict properties from electrophysiology, even in absence of full equilibrium sampling.”

It is somewhat unclear to me what the new insights exactly are:– The idea that protonation is mainly a driver of compaction, not of the gating is interesting. Unfortunately, there is no discussion of the combinatorial problem that protonation poses or mention of constant pH simulations that would be more realistic than applying fixed protonation that were proposed in the past – exploring the effect of a few selected protonation state changes would be interesting to see if the results are robust to a few perturbations.

The reviewer raises an important point regarding the effect of protonation state selection on our conclusions. We agree that it would be interesting to test different combinations of protonated residues, but the computational cost of such a combinatorial study would be many times high, and it would still suffer from the limitations of fixed (integer) protonation states similar in all subunits and during the simulations. We are also working on better constant-pH simulation methodology, in particular for interacting sites, but is unfortunately far beyond the realistic scope of this paper. However, we have added a paragraph in the Discussion section (pages 13-14) comparing our protonation protocol to two experimental studies and six simulation studies. We also agree with the reviewer regarding the potential of constant pH simulations and added a sentence and additional reference (59) to the discussion:

“Once the method has matured, constant pH simulations will likely solve the combinatorial problem that protonation state selection creates [59]”

– Discussion about desensitized state is interesting; it would have been nice to hear more from the authors about their take on the possibility to access this regime with simulations. In principle MSM have the potential to yield kinetic models that could be directly compared to the ones extracted from electrophysiology; what is the authors take on this? Will that ever be possible? When? Would a more experimental approach be desirable, instead of post-hoc comparison?

The questions about directly comparing MSMs to electrophysiology are interesting and we have added a sentence at the end of the discussion giving our opinion on this:

“Due to the possibility of direct comparison and validation to electrophysiology, these models are able to predict high-level statistical properties together with the details of channel conformational changes. This enables further exploration of changes that correlate with gating to further the fundamental understanding of pLGIC gating and, potentially, the development of state-selective drugs. We expect this methodology to be transferable to other channels with potential to predict properties from electrophysiology, even in absence of full equilibrium sampling.”